



# Application of chemical derivatization techniques combined with chemical ionization mass spectrometry to detect stabilized Criegee intermediates and peroxy radicals in the gas phase

Alexander Zaytsev[1], Martin Breitenlechner[1], Anna Novelli[2], Hendrik Fuchs[2], Daniel A. Knopf[3], Jesse H. Kroll[4], and Frank N. Keutsch[1,5,6]

[1]John A. Paulson School of Engineering and Applied Sciences, Harvard University, Cambridge, MA 02138, USA
[2]Institute of Energy and Climate Research – Troposphere (IEK-8), Forschungszentrum Jülich GmbH, 52428 Jülich, Germany
[3]School of Marine and Atmospheric Sciences, Stony Brook University, Stony Brook, NY 11794, USA
[4]Department of Civil and Environmental Engineering, Massachusetts Institute of Technology, Cambridge, MA02139, USA
[5]Department of Chemistry and Chemical Biology, Harvard University, Cambridge, MA 02138, USA
[6]Department of Earth and Planetary Sciences, Harvard University, Cambridge, MA 02138, USA

*Correspondence to*: Alexander Zaytsev (zaytsev@g.harvard.edu) and Frank N. Keutsch (keutsch@seas.harvard.edu)

**Abstract.** Short-lived highly reactive atmospheric species, such as organic peroxy radicals ($RO_2$) and stabilized Criegee intermediates (SCIs), play an important role in controlling the oxidative removal and transformation of many natural and anthropogenic trace gases in the atmosphere. Direct speciated measurements of these components are extremely helpful for understanding their atmospheric fate and impact. We describe the development of an online method for measurements of SCIs and $RO_2$ in laboratory experiments using chemical derivatization and spin trapping techniques combined with $H_3O^+$ and $NH_4^+$ chemical ionization mass spectrometry (CIMS). Using chemical derivatization agents with low proton affinity, such as electron-poor carbonyls, we scavenge all SCIs produced from a wide range of alkenes without depleting CIMS reagent ions. Comparison between our measurements and results from numeric modelling, using a modified version of the Master Chemical Mechanism, shows that the method can be used for quantification of SCIs in laboratory experiments with detection limit of $1.4 \times 10^7$ molecule cm$^{-3}$ for 30 s integration time with the instrumentation used in this study. We show that spin traps are highly reactive towards atmospheric radicals and form stable adducts with them by studying the gas-phase kinetics of their reaction with hydroxyl radical (OH). We also demonstrate that spin trap adducts with SCIs and $RO_2$ can be simultaneously probed and quantified under laboratory conditions with detection limit of $1.6 \times 10^8$ molecule cm$^{-3}$ for 30 s integration time for $RO_2$ species with the instrumentation used in this study. Spin trapping prevents radical secondary reactions and cycling, which ensures that measurements are not biased by chemical interferences, and can be implemented for detecting $RO_2$ species in the ambient atmosphere.

## 1 Introduction

Earth's atmosphere is an oxidizing environment. The initial oxidation step of volatile organic compounds (VOCs) involves reaction of a parent hydrocarbon with an oxidant. The hydroxyl radical (OH) is the most important oxidant in the atmosphere,



although oxidation can be also initiated by $O_3$, $NO_3$ and Cl- or Br-atoms. Generally, reaction of VOCs with OH, $NO_3$ and Cl-atoms occurs via H-abstraction or via addition to unsaturated carbon double bonds leading to the formation of alkyl radicals. This reaction is quickly followed by $O_2$ addition resulting in the production of organic peroxy radicals ($RO_2$). In an NO-rich
environment, $RO_2$ radicals predominantly react with NO, while at lower NO concentrations reactions with the hydroperoxy radical ($HO_2$), potentially other $RO_2$, and unimolecular reactions become more important. The common tendency is the formation of closed-shell, more oxidized VOCs (OVOCs). OVOCs may have lower volatilities than the parent hydrocarbons and may partition to the particle phase, thereby contributing to secondary organic aerosol (SOA) formation. OH, $HO_2$ and $RO_2$ radicals can form a catalytic reaction cycle, which can lead to production of tropospheric ozone as a consequence of the shift
in the $NO/NO_2$ ratio to favor formation of $NO_2$. This cycle is terminated by the formation of organic hydroperoxides and nitrates, which can be viewed as reservoirs of the corresponding radicals. Overall, atmospheric radicals, especially their cycling, play an important role in the formation of SOA and tropospheric ozone, as well as in controlling atmospheric oxidation capacity.

Organic peroxy radicals can also be formed via ozonolysis of unsaturated organic compounds. Ozonolysis of alkenes results
in the formation of primary ozonides that promptly decompose to a stable carbonyl and a vibrationally excited carbonyl oxide, also known as a Criegee intermediate (CI), some of which are thermally stabilized (SCI). SCI primarily decompose or react with water vapor (Vereecken et al., 2017) but are also believed to play a role in oxidation of $SO_2$ to form $H_2SO_4$ in the tropical regions (Khan et al., 2018). *syn*-SCI can undergo a unimolecular reaction and form a vinyl hydroperoxide, which rapidly decomposes to an OH radical and a vinyl radical. This radical is in resonance with an acetonyl-type radical, which can combine
with molecular oxygen to form an $RO_2$ species (Johnson and Marston, 2008).

Measurements of atmospheric radicals and reactive intermediates, such as $RO_2$ and SCIs, are challenging because of their high reactivity towards trace gases and surfaces and rapid cycling, which may lead to potential interferences. Highly sensitive detection systems are required to determine the minute concentrations of these species, which are typically on the order of $10^8$ molecule $cm^{-3}$ for organic peroxy radicals (Fuchs et al., 2008) and are expected to be less than $10^5$ molecule $cm^{-3}$ for SCIs
(Novelli et al., 2017). With respect to $RO_2$ species, there are several field-deployable measurement techniques available for non-speciated measurements of the sum of $RO_2$. Matrix Isolation Electron Spin Resonance Spectroscopy (MIESR) is an established, but rarely used, method for field measurements (Mihelcic et al., 1985). MIESR is an offline technique with a low time resolution (~30 min), however, its main advantage is that it does not require instrument calibration. Besides MIESR, chemical amplification and conversion systems represent another class of instruments for field studies (Edwards et al., 2003;
Hornbrook et al., 2011; Cantrell et al., 1984; Wood and Charest, 2014). In these systems peroxy radicals are not measured directly but are rather converted to other radicals or closed-shell molecules (e.g., $NO_2$ or $H_2SO_4$). A detection limit of $10^7$ molecule $cm^{-3}$ can be achieved at a temporal resolution of 15 s, however, discrimination of different $RO_2$ species is not possible (Edwards et al., 2003). In addition, secondary chemistry, i.e., additional sources of radical production and destruction, has to be considered, and care needs to be taken to ensure that measurements are not biased by any chemical interferences (Reiner et
al., 1997). Finally, laser-induced fluorescence (LIF) was also applied for ambient measurements of $RO_2$ radicals (Fuchs et al.,



2008). This technique is characterized by an excellent detection limit of $(2 - 7) \times 10^7$ molecule cm$^{-3}$ for an integration time of 30 s. Similarly to chemical amplifier systems, LIF does not allow for differentiation of various RO$_2$ species, however, although it is indirect and converts RO$_2$ to OH it does not have an amplification chain. Recently, novel mass spectrometric techniques using different ionization schemes to directly detect individual RO$_2$ species were developed (Hansel et al., 2018;

Berndt et al., 2018; Berndt et al., 2019; Nozière and Vereecken, 2019).

As for SCIs, indirect measurement techniques have been widely used. In these techniques SCIs are chemically converted to other species (e.g., H$_2$SO$_4$ or hydroxymethyl hydroperoxide, HMHP) (Berndt et al., 2014; Sipilä et al., 2014; Neeb et al., 1997). In 2008, the simplest SCI, CH$_2$OO, was directly detected for the first time (Taatjes et al., 2008). Later, synchrotron photoionization mass spectrometry was combined with the CI generation technique using diiodoalkane photolysis (Welz et al.,

2012), which spurred several studies to examine kinetics of bimolecular and unimolecular SCI reactions (Taatjes et al., 2012; Lewis et al., 2015; Chhantyal-Pun et al., 2016). Recently two new techniques for direct measurements of SCIs using Fourier transform microwave spectroscopy and chemical ionization mass spectrometry (CIMS) were introduced (Womack et al., 2015; Berndt et al., 2017).

Despite the abundance of different analytical methods used for detection of atmospheric radicals and reactive intermediates,

there is still a need for an online, direct, field-deployable technique for measuring these short-lived highly reactive compounds in a speciated way. Free radicals have been conventionally detected by chemical derivatization (CD) techniques including spin trapping in condensed-phase biological and chemical systems (Hawkins and Davies, 2014; Nosaka and Nosaka, 2017). Non-radical spin traps (e.g., nitrone spin traps) are known to react with free radicals to form stable radical adducts that can be detected with electron paramagnetic resonance spectroscopy (Roberts et al., 2016). In addition, radical spin traps (e.g.,

nitroxide radicals) are also highly reactive towards radical species such as C-centered radicals and form closed-shell adducts with them (Bagryanskaya and Marque, 2014). However, there are only few studies in which these techniques were applied for probing atmospheric radicals and intermediates. Watanabe et al. (1982) presented an offline method to quantify hydroxyl radicals using the spin trap $\alpha$-4-pyridyl-$N$-$tert$-butylnitrone $\alpha$-1-oxide (4-POBN) where condensed-phase stable adducts were detected by electron spin resonance. Recently, Giorio et al. (2017) used the spin trap 5,5-Dimethyl-1-pyrroline $N$-oxide

(DMPO) to characterize SCIs by detecting gas-phase spin trap adducts with online mass spectrometry.

Here we explore three types of CD agents, including two spin trapping agents, and show how they can be used for detection and quantification of various atmospheric radicals and reactive intermediates (Fig. 1). First, we implement the CD agent hexafluoroacetone (HFA) to characterize a wide range of gas-phase SCIs. HFA is selectively reactive towards SCIs (i.e., it is unreactive towards OH, HO$_2$ and RO$_2$), forms stable secondary ozonides with them, and has high vapor pressure and low

proton affinity (Fig. 1). Next, we use the radical spin trap (2,2,6,6-Tetramethylpiperidin-1-yl)oxyl (TEMPO) to demonstrate that spin traps are highly reactive towards radicals in the gas phase, by studying kinetics of TEMPO+OH reaction, and therefore can effectively scavenge atmospheric radicals. Finally, we utilize the non-radical spin trap DMPO to simultaneously detect atmospheric gas-phase radicals and intermediates, including SCIs and RO$_2$ species (Fig. 1). Spin trap adducts and secondary ozonides with CD agents are observed and quantified using H$_3$O$^+$ and NH$_4^+$ CIMS, which allows for speciated online



measurements of stabilized Criegee intermediates and speciated $RO_2$ radicals formed via ozonolysis of a wide range of parent
hydrocarbons. The analytical methods presented here can be used for quantification of speciated SCIs and $RO_2$ formed in
laboratory experiments as well as for field measurements.

## 2 Methods

### 2.1 Ozonolysis experiments with chemical derivatization agent HFA

The ozonolysis experiments of multiple hydrocarbons including tetramethylethylene (TME), isoprene, pentene, hexene, $\alpha$-
pinene and limonene were conducted in a flow tube reactor at ambient pressure and temperature (~290 K) and dry conditions
(relative humidity < 2%). The experimental setup consisted of a flow reactor system with a residence time of ~10 s. The parent
hydrocarbon was mixed with ozone and the chemical derivatization agent HFA ($C_3F_6O$) in the flow reactor leading to the
formation of SCIs and their scavenging as SCI·HFA adducts. SCIs are known to be highly reactive towards ketones, especially

electron poor ones such as HFA (Horie et al., 1999; Drozd et al., 2011; Drozd and Donahue, 2011; Taatjes et al., 2012). The
other advantage of employing this chemical derivatization agent is its relatively low proton affinity (PA 670.4 kJ/mol; Hunter
and Lias, 1998). Since the PA of HFA is lower than that of water, HFA cannot be protonated in $H_3O^+$ CIMS. Hence, one can
introduce significant amount of HFA to the system to make sure that all SCIs are scavenged very rapidly without any concern
that $H_3O^+$ reagent ions would be depleted. The parent hydrocarbon was vaporized from a flask filled with pure substance by

passing a constant flow of zero air regulated via a 0.1-10 $cm^3 min^{-1}$ mass flow controller (Bronkhorst). HFA flow was regulated
by another mass flow controller (Bronkhorst). Ozone was produced by passing zero air through an ozone generator using a
low-pressure mercury ultraviolet lamp. Ozone concentration was measured using an ozone monitor (2B Technologies).
A proton-transfer-reaction mass spectrometer (PTR-8000, IONICON Analytik) was used to observe formed SCI·HFA adducts
as well as parent hydrocarbons and their oxidation products. This instrument was operated using $H_3O^+$ reagent ions ($H_3O^+$

CIMS) and was directly calibrated to 10 VOCs with different functional groups (Isaacman-VanWertz et al., 2017; Isaacman-
VanWertz et al., 2018).

### 2.2 Experiments with spin traps

### 2.2.1 Kinetics experiments with spin trap TEMPO

Highly reactive spin traps are needed for effective derivatization of radicals and reactive intermediates in the gas phase. A set

of experiments, in which the reaction rate coefficient between the spin trap TEMPO ($C_9H_{18}NO$) and OH was measured, was
conducted in a flow tube experimental setup at Forschungszentrum Jülich. TEMPO is commonly used to detect carbon-
centered radicals in chemical and biological systems (Bagryanskaya and Marque, 2014) and is known to be highly reactive
towards OH in the aqueous phase (Samuni et al., 2002). TEMPO was introduced in the flow tube setup using a liquid calibration
unit (LCU, IONICON Analytik). The LCU quantitatively evaporates aqueous standards into the gas stream. TEMPO standard





was prepared gravimetrically with aqueous volume mixing ratio of 485 parts per million (ppm). A known amount (up to 10 $\mu$L min$^{-1}$) of this solution was then evaporated into a humidified gas stream of synthetic air (31 SLPM), resulting in the gas-phase TEMPO concentration of up to $4.5 \times 10^{11}$ molecule cm$^{-3}$. One part of the setup outflow was drawn to a laser photolysis – laser-induced fluorescence (LP-LIF) instrument (Lou et al., 2010), with which OH reactivity of TEMPO was measured. Laser flash photolysis of ozone was used to produce OH in the experimental setup, while LIF was applied to monitor the time

dependent OH decay. Another part of the outflow was drawn to a CIMS instrument PTR3 (IONICON Analytik) to monitor concentrations of TEMPO and its oxidation products. This instrument was operated in two ionization modes: using $H_3O^+\cdot(H_2O)_n$, n = 0–1 (as $H_3O^+$ CIMS; Breitenlechner et al., 2017) and $NH_4^+\cdot(H_2O)_n$, n = 0–2 (as $NH_4^+$ CIMS; Zaytsev et al., 2019) reagent ions. The PTR3 is designed to minimize inlet losses of sampled compounds. It was directly calibrated to 10 VOCs with different functional groups using LCU. Collision-dissociation methods were used to constrain sensitivities of $NH_4^+$

CIMS to compounds that cannot be calibrated directly (Zaytsev et al., 2019). Sensitivities were calculated in normalized duty-cycle-corrected counts per second per part per billion by volume (ndcps ppb$^{-1}$; the duty-cycle correction was done to the reference $m/z = 100$; ion signals were normalized to the primary ion signal of $10^6$ dcps).

### 2.2.2 Ozonolysis experiments with spin trap DMPO

    An additional set of ozonolysis experiments of several hydrocarbons including TME and $\alpha$-pinene were conducted in a double

flow reactor setup (Fig. 2). The goal of these experiments was to examine how spin traps can be used for simultaneous detection of stabilized Criegee intermediates and peroxy radicals. The parent hydrocarbon was mixed with ozone in the first flow tube, while the spin trap DMPO ($C_6H_{11}NO$) was introduced in the second flow tube using an LCU. A known amount (up to 10 $\mu$L min$^{-1}$) of the DMPO solution was evaporated into a humidified gas stream of synthetic air (5.4-7 SLPM), resulting in the gas-phase DMPO concentration of up to $1.1 \times 10^{13}$ molecule cm$^{-3}$. DMPO represents a class of non-radical spin traps and is

widely used to detect oxygen-centered radicals, such as OH, $HO_2$ and $RO_2$, in chemical and biological systems (Roberts et al., 2016; Van Der Zee at al., 1996). Recently, DMPO was also employed to detect SCIs in the gas phase (Giorio et al., 2017). Similar to the previous ozonolysis experiments described in Sect. 2.1, the parent hydrocarbon was vaporized from a flask filled with pure substance by passing zero air regulated by a mass flow controller, and ozone was generated using a low-pressure mercury ultraviolet lamp. The PTR3 was implemented to detect spin trap adducts with SCIs and $RO_2$ species, while ozone

levels were observed using an ozone monitor (2B Technologies).

### 2.3 Kinetic model and quantum-chemical calculations

    The Framework for 0-D Atmospheric Modelling v3.1 (F0AM; Wolfe et al., 2016) containing reactions from the Master Chemical Mechanism (MCM v3.3.1) (Jenkin et al., 1997; Saunders et al., 2003) was used to simulate photooxidation of studied alkenes in the flow reactor system and to compare the modeled concentrations of the products with the measurements. Model

calculations were constrained using physical parameters of the experimental setup (pressure and temperature) as well as to observed concentrations of the parent hydrocarbon, ozone and the chemical derivatization agent.



In order to estimate proton affinities of SCI·HFA adducts, we performed geometry optimization and proton affinity calculations with the Gaussian 09 package (Frisch et al., 2009) using the B3LYP functional (Stephens et al., 1994) and TZVP basis sets.

# 3 Results and discussion

## 3.1 Detection of speciated stabilized Criegee intermediates using chemical derivatization techniques

The primary goal of the first set of experiments was detection of speciated stabilized Criegee intermediates as adducts with the chemical derivatization agent HFA to prevent secondary reactions within the experimental setup. Starting with $(CH_3)_2COO$, an SCI produced via ozonolysis of TME, we tested the formation of SCI·HFA adducts under different experimental conditions (Fig. 3). $(CH_3)_2COO·HFA$ ($C_6H_6O_3F_6·H^+$, $m/z$ 241.03) can be easily identified in the mass-spectrum due to its unique mass defect associated with six F-atoms (Fig. S2). SCI·HFA adducts were observed when TME, ozone, and HFA were present in the experimental setup. Ozonolysis of TME also results in the formation of acetone ($C_3H_6O·H^+$, $m/z$ 59.05), which was detected in the presence of TME and ozone and was not affected by HFA addition (Fig. 3). Since the reaction rate constant of SCI·HFA with $H_3O^+$ ions is unknown, we assumed that all SCI·HFA adducts were ionized via proton transfer from hydronium ions and therefore used the sensitivity we obtained from acetone calibration to quantify detected SCI·HFA species. In addition, we did not take into account possible fragmentation of SCI·HFA adducts which may impede their detection, although a first bond cleavage would likely only break the ozonide ring structure without loss of mass. These assumptions may lead to measurement uncertainties as discussed later in this section.

We measured the $(CH_3)_2COO·HFA$ adduct signal as a function of different reactant conditions: initial TME concentration were in the range of $(1.48 - 1.85) \times 10^{11}$ molecule cm⁻³, ozone, $(6.77 - 108.2) \times 10^{12}$ molecule cm⁻³, HFA $(1.17 - 6.13) \times 10^{15}$ molecule cm⁻³. The measurements are compared to the predictions of the kinetic model in Fig. 4. Concentrations of $(CH_3)_2COO$ species were calculated using the MCM with updated kinetics data from the literature (Newland et al., 2015; Chhantyal-Pun et al., 2016; Long et al., 2018). For more details see the Supplement.

In the presence of HFA, SCI can react with HFA and form stable adducts:

$$(CH_3)_2COO + HFA \xrightarrow{k_1} (CH_3)_2COO·HFA \tag{R1}$$

The reaction rate coefficient $k_1$ was not measured experimentally, and we used the $k$-value for $CH_2OO + HFA$ reaction: $k_1 = 3 \times 10^{-11}$ cm³ molecule⁻¹ s⁻¹ (Taatjes et al., 2012). It has been suggested that the reaction between HFA and acetone oxide may be slower compared to the $CH_2OO$ one (Murray et al., 1965; Taatjes et al., 2012). However, the concentration of HFA was two orders of magnitude higher than concentrations of other chemical compounds, so even at lower $k$-values reaction with HFA remains the major chemical loss pathway for $(CH_3)_2COO$ (Fig. S3).

Observed concentrations of $(CH_3)_2COO·HFA$ agree to within a factor of 3 with concentrations predicted by the kinetic model (Fig. 4). This discrepancy can be explained by a combination of the following factors: (1) wall losses of $(CH_3)_2COO·HFA$ in the experimental setup and the PTR 8000 instrument; (2) uncertainty in the sensitivity at which the SCI·HFA adducts were



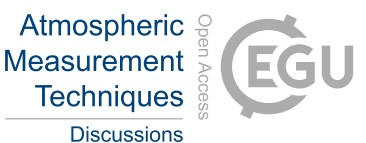

detected; (3) potential ion fragmentation of protonated SCI·HFA adducts; and (4) uncertainty in the SCI yield, and unimolecular and bimolecular reaction rate coefficients used in the kinetic model. The detection limit for $(CH_3)_2COO·HFA$ adducts was $1.4 \times 10^7$ molecule cm$^{-3}$ and was calculated for 30 s integration time as 3 standard deviations of measured background divided by derived sensitivity.

Besides TME, we also observed formation of SCI·HFA for a series of precursors including isoprene, pentene, and hexene. (Figs. S4-S6). Proton affinities (PAs) of different CI·HFA adducts were calculated using DFT methods (Table 1). A variety of these adducts can be detected using $H_3O^+$ CIMS since their PAs are significantly higher than that of water which is in agreement with experimental data (Figs. S4-S6). $CH_2OO·HFA$ cannot be detected because of its low PA (Table 1). We also did not observe SCI·HFA adducts for larger $C_{10}$ SCIs produced via ozonolysis of $\alpha$-pinene and limonene. This can be explained by the lower reactivity of larger SCIs with HFA, potential instability of these secondary ozonides in the gas phase, or their gas-wall partitioning in tubing and inside the PTR-8000 instrument.

## 3.2 Reactivity of spin traps with OH

Spin traps have been shown to be highly reactive towards free radicals and efficiently form adducts with them in the aqueous phase. However, their reactivity with atmospheric radicals and stability of formed adducts in the gas phase remain largely unknown. In order to address these questions, we conducted a set of experiments to estimate the reaction rate between the spin trap TEMPO and the hydroxyl radical by measuring its OH reactivity.

OH reactivity of a specific reactant can be calculated as a product of the reactant concentration and its reaction rate with OH (Fuchs et al., 2017):

$$k_{OH} = k_{OH+TEMPO} \cdot [TEMPO] \tag{1}$$

$k_{OH}$ was measured as a function of TEMPO concentration by varying the amount of TEMPO introduced in the experimental setup using the LCU (Fig. 5). The slope of the fitted line in Fig. 5 determines the reaction rate coefficient $k_{OH+TEMPO} = (9.3 \pm 0.9) \times 10^{-11}$ cm$^3$ molecule$^{-1}$ s$^{-1}$. This rate constant is close to the collisional limit of typical radical-molecule reactions in the atmosphere and is one order of magnitude greater than the rate constant for the same reaction in the aqueous phase ($k_{aqueous} = 7.5 \times 10^{-12}$ cm$^3$ molecule$^{-1}$ s$^{-1}$; Samuni et al., 2002). This demonstrates that TEMPO is highly reactive towards OH in the gas phase, emphasizing the applicability of spin trapping for atmospheric measurements. Furthermore, TEMPO + OH reaction leads to the formation of stable TEMPO·OH adducts that can be detected by $H_3O^+$ CIMS ($C_9H_{18}NO·H^+$, $m/z$ 174.149) and therefore could be used for quantification of hydroxyl radicals in the atmosphere (Fig. S7). Further tests are needed to compare the measurement capability of this method (e.g., sensitivity, wall losses, and potential interferences) with that of a well-established technique, such as LIF.



### 3.3 Simultaneous detection of SCIs and RO₂ species from ozonolysis of alkenes using spin trapping techniques

Next, we implemented spin trapping for detection of speciated SCIs and RO₂ species formed via ozonolysis of alkenes, starting with TME. Decomposition of the TME primary ozonide leads to formation of acetone oxide $(CH_3)_2COO$. This SCI can further

undergo a unimolecular reaction followed by $O_2$ addition to form a peroxy radical $CH_3C(=O)CH_2OO^{\cdot}$ and OH (Fig. 1). In order to detect SCIs and RO₂ species produced via ozonolysis of TME, we used a measurement method based on stabilization of these species using the spin trap DMPO followed by detection by $NH_4^+$ and $H_3O^+$ CIMS. DMPO was shown to form stable secondary ozonides with SCIs in the gas phase (Giorio et al., 2017):

$$(CH_3)_2COO + DMPO \rightarrow (CH_3)_2COO \cdot DMPO \qquad\qquad (R2)$$

DMPO is shown to be highly reactive towards SCIs ($k_{CI+DMPO} \geq 6 \times 10^{-11}$ cm³ molecule⁻¹ s⁻¹; see the Supplement for more details).

In addition, DMPO is known to be highly reactive towards oxygen-centered radicals, such as RO₂, and form stable radical adducts with them (Fig. 1):

$$CH_3C(=O)CH_2OO^{\cdot} + DMPO \rightarrow [CH_3C(=O)CH_2OO \cdot DMPO]^{\cdot} \qquad\qquad (R3)$$

We observed the formation of SCI·DMPO and RO₂·DMPO adducts both in $NH_4^+$ CIMS (e.g., $C_9H_{17}NO_3 \cdot NH_4^+$, *m/z* 205.155 and $C_9H_{16}NO_4 \cdot NH_4^+$, *m/z* 220.142) and $H_3O^+$ CIMS (e.g., $C_9H_{17}NO_3 \cdot H^+$, *m/z* 188.128 and $C_9H_{16}NO_4 \cdot H^+$, *m/z* 203.116) under different experimental conditions (Figs. 6 and S8). SCI·DMPO and RO₂·DMPO were only detected when TME, ozone, and DMPO were present in the experimental setup. Acetone, also formed via ozonolysis of TME, was observed in the presence of TME and ozone and was not affected by addition of DMPO (Figs. 6 and S8). One of the benefits of $NH_4^+$ CIMS is the

possibility of quantifying compounds for which authentic standards are not available, using a voltage scanning procedure based on collision-induced dissociation (Zaytsev et al., 2019). Based on this method, DMPO adducts with SCIs and RO₂ were detected at high sensitivities: 2,400 ndcps ppbv⁻¹ for SCI·DMPO and 2,000 ndcps ppbv⁻¹ for RO₂·DMPO (Table S1). Sensitivities were experimentally determined in each ozonolysis experiment and depend on the operational conditions of the PTR3 instrument. Detection limits for SCI·DMPO and RO₂·DMPO adducts were $3.4 \times 10^7$ and $1.6 \times 10^8$ molecule cm⁻³,

respectively. These limits of detection are calculated for 30 s integration time as 3 standard deviations of measured background divided by derived sensitivity.

In addition, we compare measured concentrations of RO₂·DMPO adducts with the concentrations predicted by the kinetic model (Fig. 7). The observed values are an order of magnitude lower than the modeled ones. Several factors can contribute to this discrepancy: (1) gas-wall partitioning of RO₂ species and RO₂·DMPO adducts in the flow tube setup and inside the PTR3

instrument; (2) uncertainty in sensitivity at which RO₂·DMPO adducts were detected; (3) potential fragmentation of RO₂·DMPO; and (4) uncertainties in the reaction rate coefficient $k_{RO_2+DMPO}$. In our model we assumed that the major fraction of RO₂ species was scavenged by DMPO. This assumption is valid if $k_{RO_2+DMPO}$ is larger than $1 \times 10^{-12}$ cm³ molecule⁻¹ s⁻¹. Otherwise, other loss channels for peroxy radicals, especially the RO₂+RO₂ reaction, become more important (Fig. S9).





Additional experiments under different conditions and intercomparison with established methods (i.e., LIF) are needed to
further estimate the measurement capability of the proposed analytical method.

Finally, we employed spin trapping for detection of SCIs and organic peroxy radicals formed via ozonolysis of larger cyclic
alkenes, such as $\alpha$-pinene. Decomposition of the $\alpha$-pinene primary ozonide yields four different $C_{10}$-SCIs, all of which have
the same molecular formula $C_{10}H_{16}O_3$ (Newland et al., 2018). These SCIs can further isomerize to form primary peroxy radicals
$C_{10}H_{15}O_4$ and OH. Autoxidation of $C_{10}H_{15}O_4$-$RO_2$ species can in turn result in formation of several more oxygenated peroxy
radicals $C_{10}H_{15}O_x$, x = 5-9 (Zhao et al., 2018). Signals of SCI·DMPO ($C_{10}H_{16}O_3$·DMPO) and $RO_2$·DMPO ($C_{10}H_{15}O_x$·DMPO,
x = 4-9) adducts were observed both in $NH_4^+$ and $H_3O^+$ CIMS (Figs. 8, S10, S11). This demonstrates that this analytical method
allows for simultaneous detection of a wide range of atmospheric radicals, including the ones with high oxygen content (an
O:C ration of up to 0.9) that are formed via autoxidation pathway, and can be used to study kinetics of these species in the
laboratory.

## 4 Conclusions

In summary, we experimentally demonstrated the measurement of speciated, short-lived highly reactive atmospheric
compounds, such as Criegee intermediates, organic peroxy radicals and hydroxyl radicals, formed via ozonolysis of alkenes.
The analysis was carried out using chemical derivatization techniques, including spin trapping, while the detection of formed
radical adducts and closed-shell secondary ozonides was performed by the means of $H_3O^+$ and $NH_4^+$ CIMS. Detected adducts
and secondary ozonides have unique mass defects and can therefore be clearly separated from other observed compounds in
the mass spectrum. Implementation of chemical derivatization agents with lower proton affinity allows for the full scavenging
and quantification of stabilized Criegee intermediates without depleting CIMS reagent ions. We show that spin traps can be
used to effectively scavenge atmospheric radicals and reactive intermediates by demonstrating their high reactivity with
radicals in the gas phase using the TEMPO+OH reaction as an example. Using the spin trap DMPO, SCIs and $RO_2$ species
can be simultaneously detected while quantification of observed adducts can be done without their direct calibration. The
detection limits of spin trap and chemical derivatization agent adducts of $1.4 \times 10^7$ molecule cm$^{-3}$ for SCIs and $1.6 \times 10^8$
molecule cm$^{-3}$ for $RO_2$ for 30 s integration time were estimated for the instrumentation used here and show promise that these
techniques would also work when sampling ambient air. In particular, this method fundamentally enables any CIMS instrument
to detect radicals and SCIs. Since spin traps, such as DMPO and TEMPO, are reactive towards a plethora of atmospheric
radicals and reactive intermediates, including $RO_2$, SCIs, and OH, implementation of such spin traps results in the effective
suppression of the radical secondary chemistry and, thus, elimination of potential chemical interferences. The direct method
for speciated SCIs and $RO_2$ measurements provides a means to study the atmospheric chemistry of these compounds. We stress
that the quantification of $RO_2$ species was done under well-defined laboratory conditions using the CID technique such that
the estimated sensitivities are likely unique to the electric fields, pressures and flows of the $NH_4^+$ CIMS instrument. Further



validation of the proposed analytical methods in more complex environments closer to the ambient conditions and intercomparison with established methods (i.e., LIF) are needed.

For the future application of the method in field and laboratory experiments, various modifications of the experimental setup can be implemented to improve its measurement capability. We plan to synthesize and test new chemical derivatization agents optimized for the gas-phase measurements with respect to their vapor pressure, selective reactivity and by labelling with atomic

isotopes to simplify mass spectrometric detection and improve detection limits. With labeled spin traps, the identification of reactive intermediates may be greatly simplified and detection limits may be further improved, as the spin trap can provide a unique signature in the complex mass-spectrum and move the observed $m/z$ to a region with very low background.

*Acknowledgements.* This work has received funding from the Harvard Global Institute. Martin Breitenlechner acknowledges

support from the Austrian science fund (FWF; grant J-3900). Hendrik Fuchs and Anna Novelli acknowledge support from the European Research Council (ERC) under the European Union's Horizon 2020 research and innovation programme (SARLEP grant agreement No. 681529). Daniel Knopf acknowledges support from the U.S. National Science Foundation (grant no. AGS-1446286). Jesse Kroll acknowledges support from the U.S. National Science Foundation (grant no. AGS-1638672).





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



**Table 1: Proton affinities (PAs) of HFA, water and secondary ozonides produced in reactions of SCIs with HFA. Species with PAs higher than that of water can be detected in $H_3O^+$ CIMS.**

| Species | PA, kcal/mol | Reference |
|---|---|---|
| $CH_2OO \cdot HFA$ | 662.9 | This work |
| HFA | 670.4 | Hunter and Lias (1998) |
| $H_2O$ | 691 | Hunter and Lias (1998) |
| $CH_3CH_2CHOO \cdot HFA$ | 720.7 | This work |
| $CH_3CH_2CH_2CHOO \cdot HFA$ | 730.8 | This work |
| $(CH_3)_2COO \cdot HFA$ | 747.2 | This work |
| $(CH_2=C(CH_3))CHOO \cdot HFA$ | 779.6 | This work |

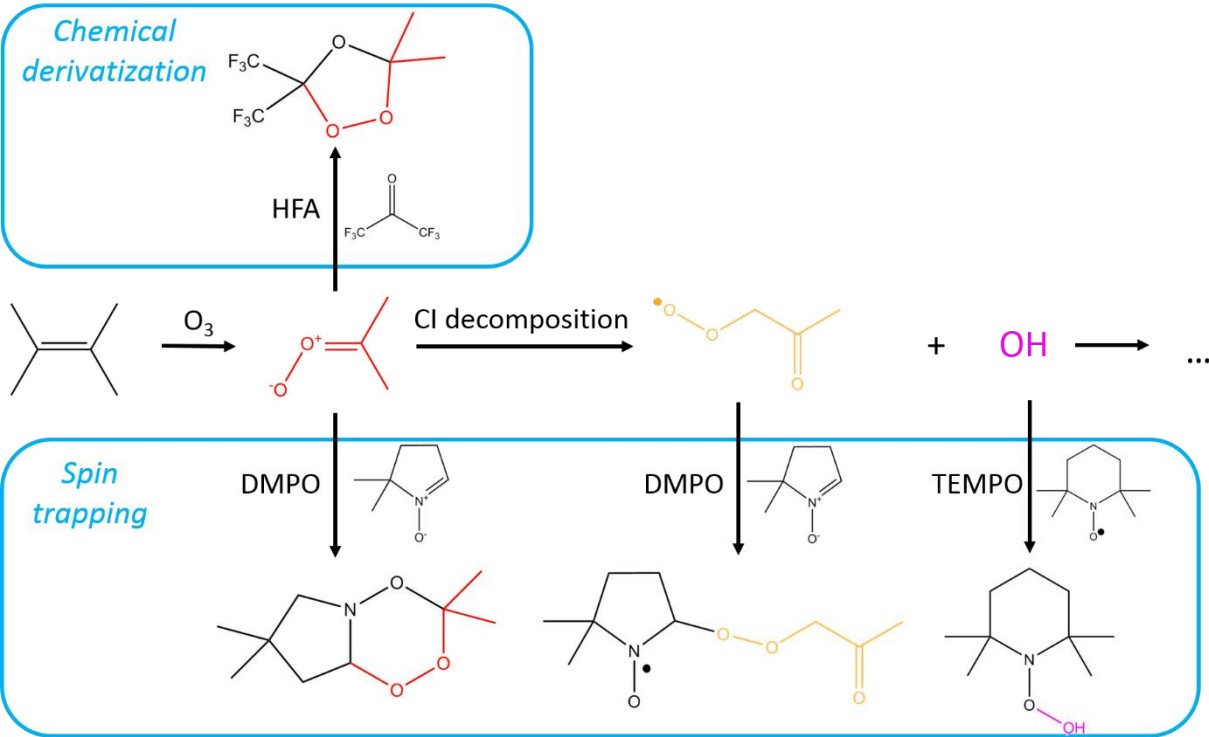

**Figure 1: Mechanism of tetramethylethylene (TME) ozonolysis. Stabilized Crigee intermediate (shown in red) can be scavenged by the chemical derivatization agent HFA or the spin trap DMPO, or decompose to peroxy radical (shown in yellow) and OH. RO₂ and OH species can in turn react with spin traps. Reactions involving SCI are from MCM v3.3.1 (Jenkin et al., 1997) and Giorio et al. (2017).**



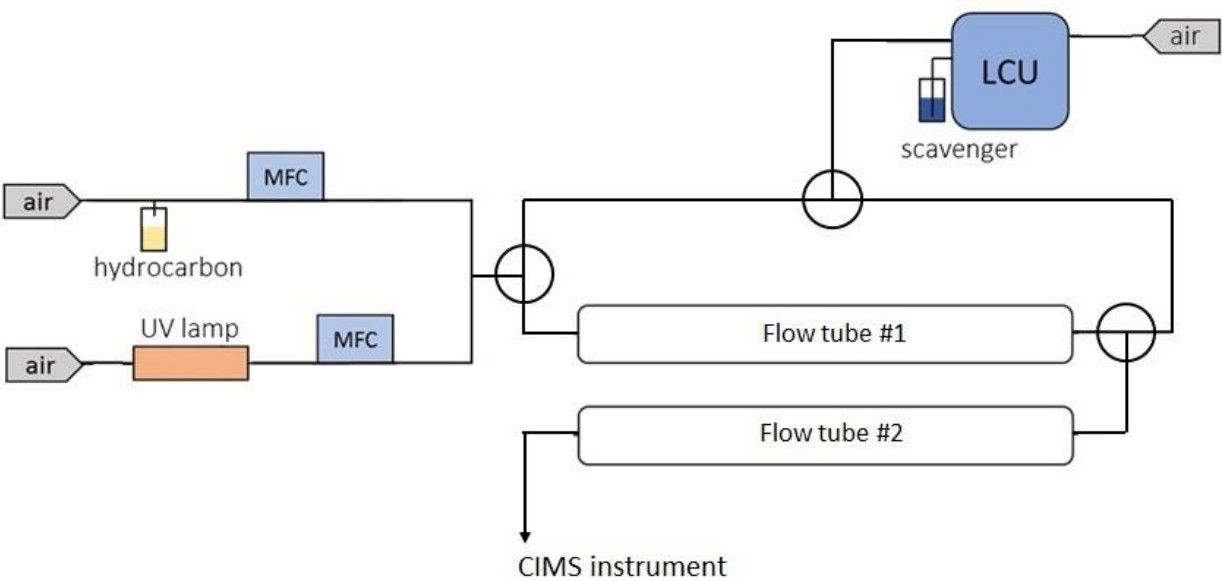


**Figure 2: Schematic of experimental setup used to detect SCIs and RO₂ with the spin trap DMPO. DMPO was introduced in the experimental setup using a liquid calibration unit (LCU, IONICON Analytik).**

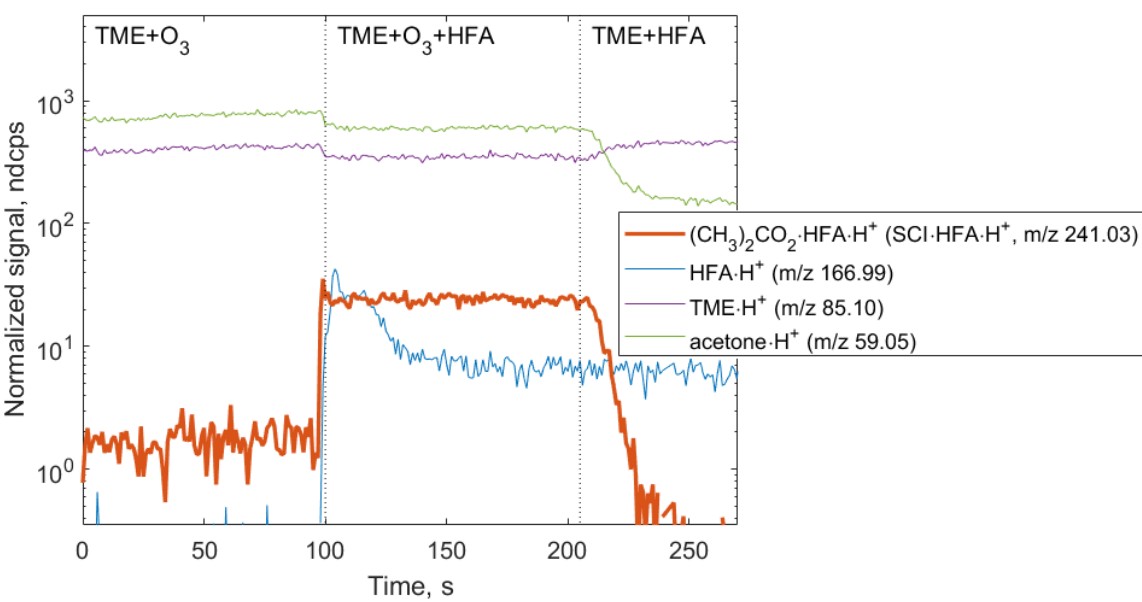

**Figure 3: Ion tracers observed in a TME ozonolysis experiment as a function of different reactant conditions. Reactant**
**concentrations are [TME] = $1.85 \times 10^{12}$; [O₃] = $1.67 \times 10^{13}$; [HFA] = $6.09 \times 10^{15}$ molecule cm⁻³. (CH₃)₂COO·HFA·H⁺ ion (red tracer, $m/z$ 241.03) is observed when TME, HFA, and O₃ are present in the system.**





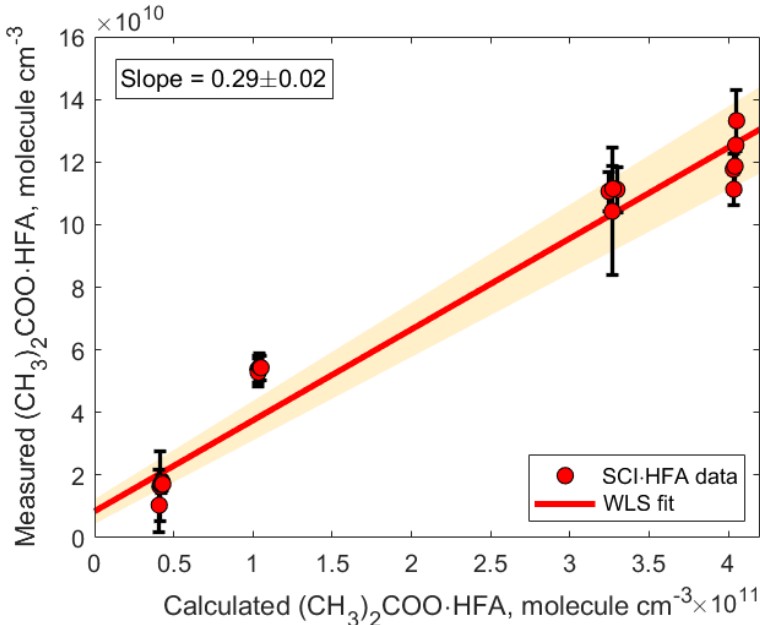

**Figure 4: Correlation plot comparing measured and calculated concentrations of (CH₃)₂COO·HFA. The adducts were detected using H₃O⁺ CIMS as (CH₃)₂COO·HFA·H⁺ (*m/z* 241.03). The slope is calculated using weighted least squares (WLS). A 95% confidence**
**interval is estimated via a Monte Carlo simulation (N=5000) and shown using red shading.**

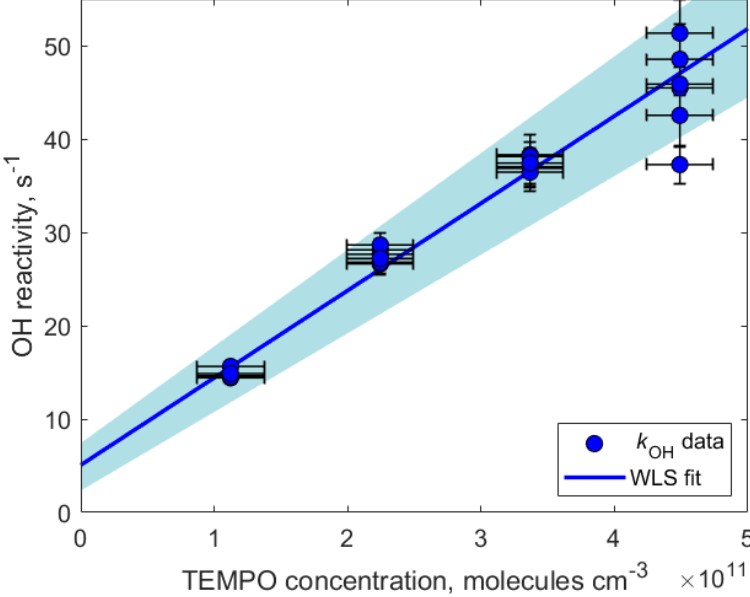

**Figure 5: OH reactivity measured as a function of TEMPO concentration. The slope determining the reaction rate coefficient**
$k_{\text{TEMPO+OH}} = (9.3 \pm 0.9) \times 10^{-11}$ **cm³ molecule⁻¹ s⁻¹ is calculated using weighted least squares (WLS). A 95% confidence interval is estimated via a Monte Carlo simulation (N=5000) and shown using blue shading. The intercept (5.1±2.4) s⁻¹ can be explained by**
**other OH reactants such as O₃, NO, NO₂, and CO.**





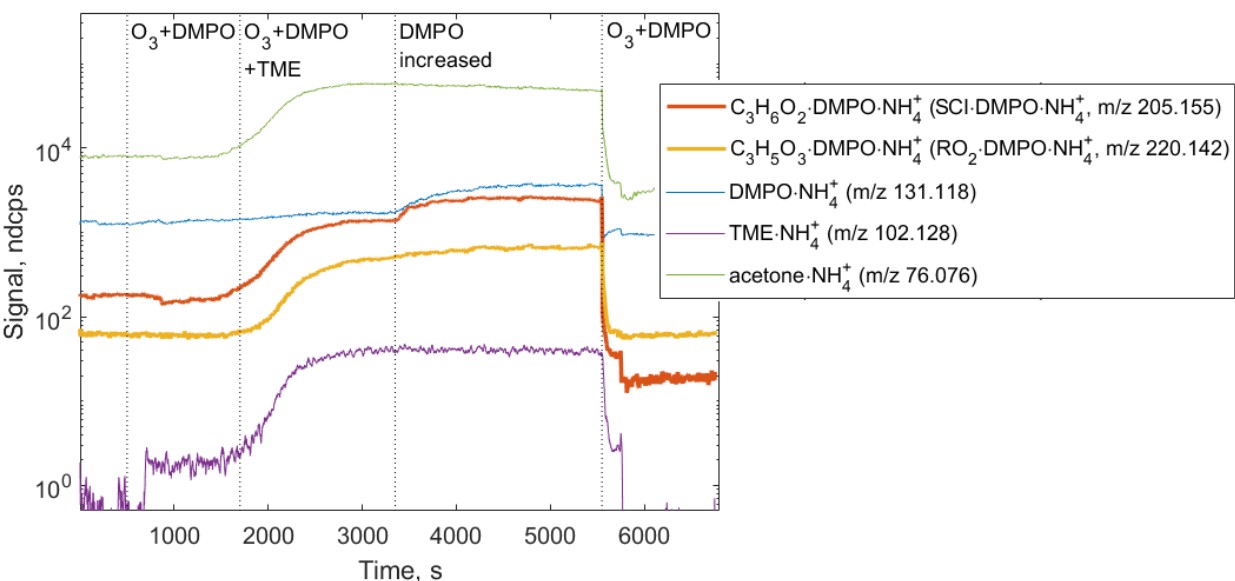

**Figure 6: Ion tracers observed by NH₄⁺ CIMS in a TME ozonolysis experiment as a function of different reactant conditions. Reactant concentrations are [TME] = $3.69 \times 10^{11}$; [O₃] = $7.87 \times 10^{12}$; [DMPO] = $2.01 \times 10^{12}$ molecule cm⁻³.**

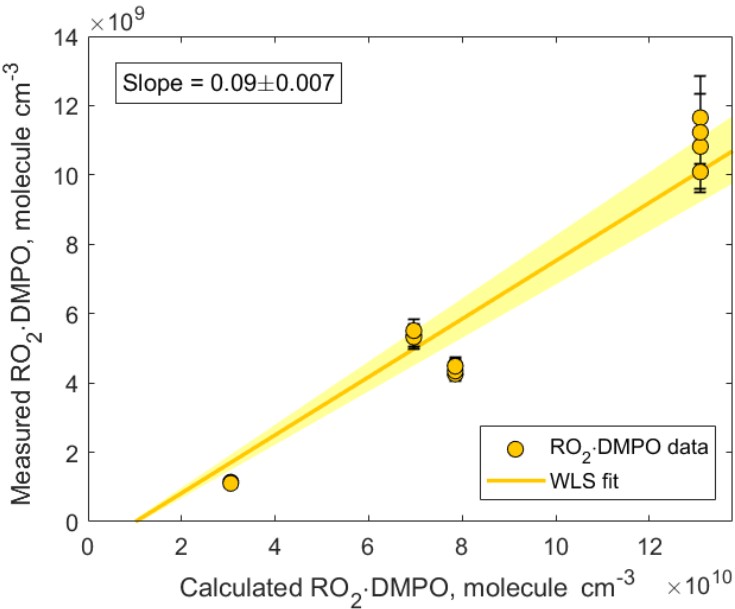

**Figure 7: Correlation plot comparing measured and calculated concentrations of CH₃C(=O)CH₂OO·DMPO. The adducts were detected using NH₄⁺ CIMS as CH₃C(=O)CH₂OO·DMPO·NH₄⁺ (*m/z* 220.142). The slope is calculated using weighted least squares (WLS). A 95% confidence interval is estimated via a Monte Carlo simulation (N=5000) and shown using yellow shading.**


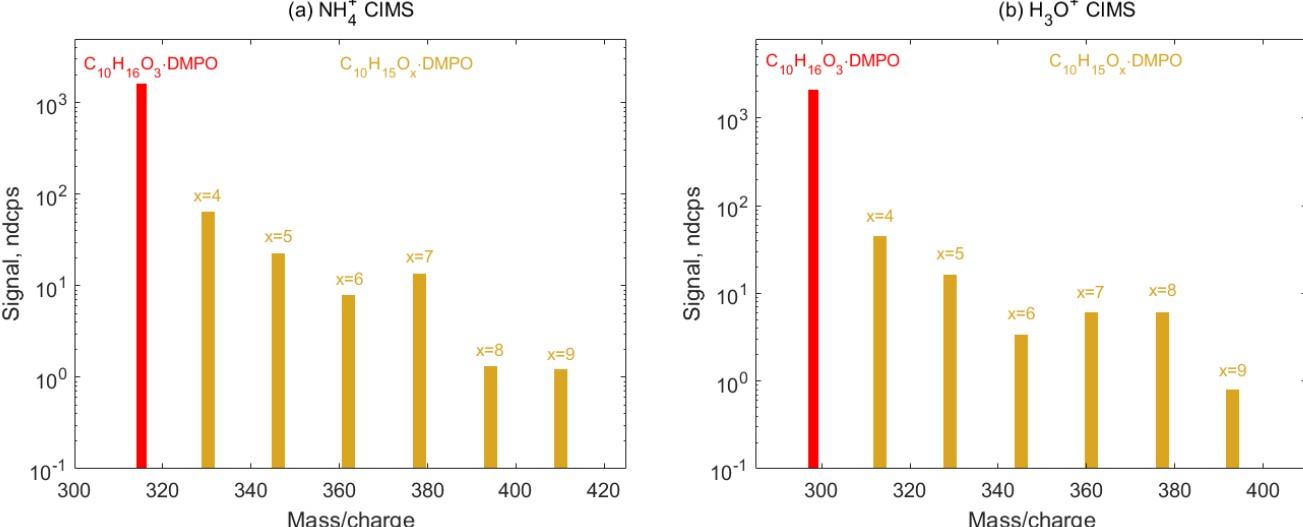

**Figure 8: Mass spectra of SCI·DMPO (red) and RO₂·DMPO (yellow) adducts in $\alpha$-pinene ozonolysis experiments observed using (a) NH₄⁺ CIMS and (b) H₃O⁺ CIMS. Primary RO₂ species ($C_{10}H_{15}O_4$) are formed via CI isomerization and can in turn undergo various autoxidation reactions resulting in formation of several organic peroxy radicals ($C_{10}H_{15}O_x$, x=5-9), which were detected bas adducts with the spin trap DMPO by the means of NH₄⁺ and H₃O⁺ CIMS.**