# Peer review of "Application of chemical derivatization techniques combined with chemical ionization mass spectrometry to detect stabilized Criegee intermediates and peroxy radicals in the gas phase"

_Atmospheric Measurement Techniques, 2020_

## Referee Comment (RC1) · Anonymous Referee #1 · 9 Nov 2020

Manuscript ID: amt-2020-335

The authors are describing experimental findings from an online method for the detection of thermalized Criegee intermediates (CI) and RO2 radicals in different laboratory setups. CIs have been observed via HFA titration or DMPO derivatization and RO2 radicals via DMPO or TEMPO derivatization. Analysis was carried out by means of a PTR3 mass spectrometer running in the H3O+ or NH4+ mode. CI detection via HFA adducts was successful in the case of the ozonolysis of TME, isoprene, pentene and hexene, but not for the expected CIs arising from the ozonolysis of selected terpenes.

[Figure]

Also the simplest CI, CH2OO, was not measurable. Examples for RO2 measurements are given from the ozonolysis (incl. OH reaction?) of TME and alpha-pinene. The stated detection limit for CIs is about 10(7) molecules/cc and that for RO2s about 10(8) molecules/cc for 30 s integration time. The topic of this paper is well suited for AMT. Some clarifications are needed before publication can be recommended.

- Line 53: Atmospheric RO2 radical concentrations in the order of 10(8) molecules/cc are not generally valid. It stands mainly for CH3O2, concentration levels of other RO2 radicals can be much lower.

- Line 104: Please provide a table with the initial reactant concentrations and the calculated amount of reacted olefin for a better understanding what has been done.

- Line 143: Also here, please state the initial reactant conditions. What was the residence time in the respective flow tubes? If I understand it right, in the first flow tube the O3(OH?) + TME/alpha-pinene reaction was running without OH scavenger and the second flow tube served for product derivatization by DMPO (but TME/alpha-pinene conversion was still running)? Please provide a more precise insight what's going on in the different parts of this flow-through experiment.

- Line 186: The Donahue group, ref: 10.1021/jp108773d, used k((CH3)2COO + HFA) = 2 x 10(-13) cc/s, about 2 orders of magnitude lower as the rate coefficient used in this work. Is the HFA concentration still high enough for complete conversion of (CH3)2COO with HFA?

- Line 197: How good is the agreement model vs. measurement in the case of the ozonolysis of isoprene, pentene and hexene?

- Line 204: What is the detection limit of OH radicals via the TEMPO derivatization as a result of this work? Giorio et al., ref:10.1021/jacs.6b10981, were not able to follow OH production from alpha-pinene ozonolysis using a similar technique. Is it really possible to measure steady-state OH in a reaction system by means of this technique?

- Line 222: I think these experiments have been done in the double flow-tube setup, right? So, you should see the resulting RO2 radicals from ozonolysis as well as those from the OH reaction if no OH scavenger is used. That means in the case of TME also the primarily formed HO-C6H12O2 radicals should be visible in addition to acetonylperoxy radicals from the ozone reaction? And in the case of alpha-pinene, HO-C10H16O2 radicals (and subsequent autoxidation products) must be there along with the ozonolyis-derived RO2s. Please comment!

Another point: Hansel et al., ref: 10.1016/j.atmosenv.2018.04.023, are stating a detection limit of 2 x 10(5) molecules/cc for RO2 radicals and closed shell products from cyclohexene ozonolysis using a similar(or same) mass spec with NH4+ ionization. That means the authors should be able to monitor the RO2 radicals directly at the outflow w/o derivatization? That could be helpful for the assessment of the derivatization procedure.

- Line 259 and fig.8: Higher oxidized RO2 radicals arising from pure autoxidation steps show a mass difference of 32 mass units due to step-by-step insertion of molecular oxygen. A mass difference of 16 mass units points to efficient bimolecular RO2 steps altering the autoxidation-governed RO2 distribution. So, as already said, it would be fine to have the complete reaction conditions to get an idea how important RO2 + RO2 could be.

---

## Referee Comment (RC2) · Anonymous Referee #3 · 23 Dec 2020

This study presents the development of an online method for measurements of SCIs and RO2 in laboratory experiments using chemical derivatization and spin trapping techniques combined with H3O+ and NH4+ chemical ionization mass spectrometry. Application of this method is demonstrated using laboratory ozonolysis experiments of multiple hydrocarbons including TME, isoprene, pentene, hexene, alpha-pinene and limonene. The detection limits of spin trap and chemical derivatization agent adducts are estimated to be 1.4E+7 molecule cm-3 for SCIs and 1.6E+8 molecule cm-3 for RO2 for 30 s integration time for the instrumentation used in this study. This manuscript is

well written and within the scope of the journal. I recommend this manuscript to be published in AMT after the following issues be addressed.

Page 6, Line 166-167: Is there any evidence for using HFA with SCIs to prevent secondary reactions?

Page 6, Line 191-194 and Page 8, Line 249-251: Could the author give more detailed explanations or quantitative analysis for these four reasons?

Page 9, Line 277-278: Since there could be various RO2 in ambient air, how does the author think about the feasibility of using the CID technique to measure ambient air?

Supplement page 8: In FigureS11, at the beginning of the period DMPO+O3, why did the SCI adduct (m/z 315.228) get a little increasing?

Page 8, Line 234 and Supplement page 2, Line7: The last two letters of the word "CH3C(=O)CH2OO" use two different fonts.

---

## Author Comment (AC1) · 9 Jan 2021

**Response to reviewer #1's comments**

Reviewer comments are in **bold**. Author responses are in plain text. Excerpts from the manuscript are in *italics*. Modifications to the manuscript are in *blue italics*. Page and line numbers in the responses correspond to those in the original AMTD paper.

**The authors are describing experimental findings from an online method for the detection of thermalized Criegee intermediates (CI) and $RO_2$ radicals in different laboratory setups. CIs have been observed via HFA titration or DMPO derivatization and $RO_2$ radicals via DMPO or TEMPO derivatization. Analysis was carried out by means of a PTR3 mass spectrometer running in the $H_3O^+$ or $NH_4^+$ mode. CI detection via HFA adducts was successful in the case of the ozonolysis of TME, isoprene, pentene and hexene, but not for the expected CIs arising from the ozonolysis of selected terpenes. Also the simplest CI, $CH_2OO$, was not measurable. Examples for $RO_2$ measurements are given from the ozonolysis (incl. OH reaction?) of TME and alpha-pinene. The stated detection limit for CIs is about $10^7$ molecules/cc and that for $RO_2$ about $10^8$ molecules/cc for 30 s integration time. The topic of this paper is well suited for AMT. Some clarifications are needed before publication can be recommended.**

We would like to thank the reviewer for the positive reception of our work and constructive comments that helped us to improve our manuscript. Below we provide our replies to the reviewer's comments. Page and line numbers in the responses correspond to those in the AMTD paper.

1. **Line 53: Atmospheric $RO_2$ radical concentrations in the order of $10^8$ molecules/cc are not generally valid. It stands mainly for $CH_3O_2$, concentration levels of other $RO_2$ radicals can be much lower.**

   We modify the following sentence by specifying ambient concentrations of $RO_2$ species (P2 L52):

   *Highly sensitive detection systems are required to determine the minute concentrations of these species, . Concentrations of the smallest organic peroxy radicals, $CH_3O_2$, are typically on the order of $10^8$ molecule cm$^{-3}$ while concentrations of other $RO_2$ species can be much lower (Fuchs et al., 2008). As for SCIs, their concentrations are expected to be less than $10^5$ molecule cm$^{-3}$ (Novelli et al., 2017).*

2. **Line 104: Please provide a table with the initial reactant concentrations and the calculated amount of reacted olefin for a better understanding what has been done.**

   We include the following table containing the initial reactant concentrations and the calculated amount of reacted olefin in the SI:

   *Table S2: Descriptions of ozonolysis experiments with HFA*

| Olefin | Initial olefin concentration, molecule cm$^{-3}$ | $O_3$ concentration, molecule cm$^{-3}$ | HFA concentration, molecule cm$^{-3}$ | Calculated amount of reacted olefin, % |
|---|---|---|---|---|

| | | | | |
|---|---|---|---|---|
| *TME* | $1.85 \cdot 10^{12}$ | $1.67 \cdot 10^{13}$ | $6.09 \cdot 10^{15}$ | 17% |
| *isoprene* | $1.23 \cdot 10^{13}$ | $3.20 \cdot 10^{14}$ | $5.35 \cdot 10^{15}$ | 6% |
| *pentene* | $4.18 \cdot 10^{13}$ | $6.15 \cdot 10^{13}$ | $5.35 \cdot 10^{15}$ | 14% |
| *hexene* | $2.21 \cdot 10^{13}$ | $2.95 \cdot 10^{14}$ | $5.35 \cdot 10^{15}$ | 50% |
| *α-pinene* | $2.70 \cdot 10^{12}$ | $3.20 \cdot 10^{14}$ | $5.35 \cdot 10^{15}$ | 37% |
| *limonene* | $2.10 \cdot 10^{12}$ | $3.45 \cdot 10^{14}$ | $5.35 \cdot 10^{15}$ | 67% |

3. **Line 143: Also here, please state the initial reactant conditions. What was the residence time in the respective flow tubes? If I understand it right, in the first flow tube the $O_3$(OH?) + TME/alpha-pinene reaction was running without OH scavenger and the second flow tube served for product derivatization by DMPO (but TME/alpha-pinene conversion was still running)? Please provide a more precise insight what's going on in the different parts of this flow-through experiment.**

We add the following discussion on the experimental setup used during the ozonolysis experiments with spin trap DMPO (P5 L146):

*Experimental setup consisted of two identical ~2.1L flow reactors. The parent hydrocarbon was mixed with ozone in the first flow  reactor with a residence time of ~28s. Similar to the previous ozonolysis experiments described in Sect. 2.1, the parent olefin was vaporized from a flask filled with pure substance by passing zero air regulated by a mass flow controller, and ozone was generated using a low-pressure mercury ultraviolet lamp.  We used an LCU to introduce the spin trap DMPO in the second flow reactor with a residence time of ~23s. A known amount (up to 10 $\mu$L min$^{-1}$) of the DMPO solution was evaporated into a humidified gas stream of synthetic air (5.4-7 SLPM), resulting in the gas-phase DMPO concentration of up to $1.1 \times 10^{13}$ molecule cm$^{-3}$. The second flow reactor served for derivatization of SCIs and $RO_2$ species by DMPO while the parent hydrocarbon was still reacting with ozone. Hence, we conducted integrated production measurements of SCIs and $RO_2$ species formed in both flow reactors. The PTR3 was used to detect  SCI·DMPO and $RO_2$·DMPO adducts, while ozone levels were observed using an ozone monitor (2B Technologies).*

In addition, we include the following table containing the initial reactant concentrations and the calculated amount of reacted olefin in the SI:

*Table S3: Descriptions of ozonolysis experiments with DMPO*

| *Olefin* | *Initial olefin concentration, molecule cm$^{-3}$* | *$O_3$ concentration, molecule cm$^{-3}$* | *DMPO concentration, molecule cm$^{-3}$* | *Calculated amount of reacted olefin, %* |
|---|---|---|---|---|
| *TME* | $3.69 \cdot 10^{11}$ | $7.87 \cdot 10^{12}$ | $2.01 \cdot 10^{12}$ | 43% |
| *α-pinene* | $4.92 \cdot 10^{11}$ | $1.03 \cdot 10^{13}$ | $1.10 \cdot 10^{13}$ | 9% |

4. **Line 186: The Donahue group, ref: 10.1021/jp108773d, used $k_{(CH_3)_2COO+HFA} = 2 \times 10^{-13}$ cc/s, about 2 orders of magnitude lower as the rate coefficient used in this work. Is the HFA concentration still high enough for complete conversion of $(CH_3)_2COO$ with HFA?**

Since the proton affinity of HFA is lower than that of water, we were able to introduce significant amounts of HFA (see Table S2 above) to make sure that HFA remains the major chemical loss even if $k_{(CH_3)_2COO+HFA} = 2 \times 10^{-13}$ molecule cm$^{-3}$ s$^{-1}$. We update Fig S3 in the SI:

[Figure]

*Figure S3: Chemical losses of stabilized Criegee intermediates (CH3)2COO calculated assuming different $k_{SCI+HFA}$ reaction rates under experimental conditions. $k_{SCI+HFA} = 3 \times 10^{-11}$ cm³ molecule$^{-1}$ s$^{-1}$ corresponds to the rate constant for CH₂OO + HFA reaction (Taatjes et al., 2012). Previous studies used lower rate constant ($2 \times 10^{-13}$ molecule cm$^{-3}$ s$^{-1}$; Drozd et al., 2011). Even at lower values of the reaction rate the major chemical loss pathway for SCI is the reaction with HFA.*

We also add the reference to the work by the Donahue group in the manuscript (P6 L187):

*It has been suggested that the reaction between HFA and acetone oxide may be slower compared to the CH₂OO one (Murray et al., 1965; Taatjes et al., 2012) while $k_{(CH_3)_2COO+HFA} = 2 \times 10^{-13}$ molecule cm$^{-3}$ s$^{-1}$ was used in the previous studies (Drozd et al., 2011).*

5. **Line 197: How good is the agreement model vs. measurement in the case of the ozonolysis of isoprene, pentene and hexene?**

In the case of the ozonolysis of isoprene, pentene and hexene, our measurements of SCI·HFA are one to two orders of magnitude lower than the model prediction. There are several factors that can contribute to this discrepancy:
1. Yields of SCIs for larger intermediates might be off. MCM assumes the same yield of 0.18 for CH₃CHOO, CH₃CH₂CHOO and CH₃CH₂CH₂CHOO, however, measured yields of these intermediates vary by up to a factor of 2 (Newland et al., 2015 and references therein). In

addition, some studies suggested that yields of larger SCIs (e.g., $C_4$-SCI) are significantly smaller than that of $CH_2OO$ (Nguyen et al., 2016).

2. Unimolecular decomposition of SCI is not taken into account in the model. MCM includes only bimolecular loss reactions for $CH_3CHOO$, $CH_3CH_2CHOO$ and $CH_3CH_2CH_2CHOO$, while some studies suggest that SCI unimolecular rates increase with size and become more important (Nguyen et al., 2016; Newland et al., 2015).

3. The reaction rate coefficient between larger SCI and the derivatization agent HFA is unknown. As the reviewer pointed out earlier, the reaction rate coefficient is expected to be lower for larger SCIs, but it has not been measured directly. While we introduced significant amounts of HFA in the experimental system to ensure that the reaction with HFA remains the major chemical loss for SCIs, we cannot be certain that all SCIs were scavenged by HFA.

Based on these factors and associated uncertainties in both the model and measurements, we think that presenting the model vs. measurement agreement for isoprene, pentene, and hexene falls beyond the scope of this study.

6. **Line 204: What is the detection limit of OH radicals via the TEMPO derivatization as a result of this work? Giorio et al., ref:10.1021/jacs.6b10981, were not able to follow OH production from alpha-pinene ozonolysis using a similar technique. Is it really possible to measure steady-state OH in a reaction system by means of this technique?**

We estimate the detection limit of OH radicals via the TEMPO derivatization for our setup to be $\sim 6 \times 10^6$ molecule $cm^{-3}$. This limit of detection is calculated for a 1 s integration time of TEMPO·OH signal as three standard deviations of measured background divided by derived sensitivity for TEMPO. The purpose of TEMPO derivatization experiments was to demonstrate that chemical derivatization agents, including spin traps, are highly reactive towards atmospheric radicals and reactive intermediates rather than to fully describe this method to detect OH radicals. As we state in the manuscript (P7 L219), further tests are required to compare the measurement capability of this method with that of a well-established technique, such as LIF. Whether steady-state OH concentration can be measured will depend on the experimental setup and what averaging time is acceptable. For example, with 10 min averaging the detection limit can be reduced to $2.5 \times 10^5$ molecule $cm^{-3}$, which is in a useful range. Furthermore, other CIMS instruments have achieved lower detection limits. Thus, we believe detection of OH is feasible, depending on conditions and instrumentation. While we agree with the reviewer that it would be interesting to check if it would be possible to observe OH from $\alpha$-pinene ozonolysis, we think that conducting such experiments lies beyond the scope of this manuscript.

7. **Line 222: I think these experiments have been done in the double flow-tube setup, right? So, you should see the resulting $RO_2$ radicals from ozonolysis as well as those from the OH reaction if no OH scavenger is used. That means in the case of TME also the primarily formed HO-$C_6H_{12}O_2$ radicals should be visible in addition to acetonylperoxy radicals from the ozone reaction? And in the case of alpha-pinene, HO-$C_{10}H_{16}O_2$ radicals (and subsequent autoxidation products) must be there along with the ozonolyis-derived $RO_2$s. Please comment!**

We observed formation of RO$_2$ species formed via OH-oxidation of TME. We include the following discussion (P8 L239) and edit Fig. 6 by adding the corresponding tracer to it:

*OH radicals, formed via decomposition of SCI, can in turn react with TME and lead to formation of another RO$_2$ species OH-C$_6$H$_{12}$OO$^\cdot$. This radical was detected as the C$_6$H$_{13}$O$_3$·DMPO adduct (C$_{12}$H$_{24}$NO$_4$, m/z 264.205; Fig. 6).*

[Figure]

*Figure 6: Ion tracers observed by NH$_4^+$ CIMS in a TME ozonolysis experiment as a function of different reactant conditions. Reactant concentrations are [TME] = $3.69 \times 10^{11}$; [O$_3$] = $7.87 \times 10^{12}$; [DMPO] = $2.01 \times 10^{12}$ molecule cm$^{-3}$.*

In addition, we also observed formation of HO-C$_{10}$H$_{16}$O$_2$ species and subsequent autooxidation products in the case of $\alpha$-pinene. We include the following discussion (P9 L261) and edit Figs. S10 and S11:

*OH radicals, formed via decomposition of SCI, can in turn react with $\alpha$-pinene and lead to formation of OH-derived RO$_2$ species $C_{10}H_{17}O_3$ and subsequent autooxidation RO$_2$ species $C_{10}H_{17}O_5$ (Berndt et al., 2016). These radicals were detected as the RO$_2$·DMPO adducts (Figs. S10 and S11).*

[Figure]

*Figure S10: Ion tracers observed by $H_3O^+$ CIMS in an $\alpha$-pinene ozonolysis experiment as a function of different reactant conditions. Reactant concentrations are [$\alpha$-pinene] = $4.92 \times 10^{11}$; [$O_3$] = $1.03 \times 10^{13}$; [DMPO] = $1.10 \times 10^{13}$ molecule cm$^{-3}$.*

[Figure]

*Figure S11: Ion tracers observed by $NH_4^+$ CIMS in an $\alpha$-pinene ozonolysis experiment as a function of different reactant conditions. Reactant concentrations are [$\alpha$-pinene] = $4.92 \times 10^{11}$; [$O_3$] = $1.03 \times 10^{13}$; [DMPO] = $1.10 \times 10^{13}$ molecule cm$^{-3}$.*

8. **Another point: Hansel et al., ref: 10.1016/j.atmosenv.2018.04.023, are stating a detection limit of $2 \times 10^5$ molecules/cc for $RO_2$ radicals and closed shell products from cyclohexene ozonolysis using a similar (or same) mass spec with $NH_4^+$ ionization. That means the authors should be able to monitor the $RO_2$ radicals directly at the outflow w/o derivatization? That could be helpful for the assessment of the derivatization procedure.**

We agree with the reviewer that it would be interesting to conduct simultaneous measurements of $RO_2$ species with and without using derivatization agents, however, our setup was not designed for this type of experiments. In addition, there are several disadvantages associated with direct

measurements of $RO_2$ species: (1) potential interferences from secondary chemistry, i.e., additional sources or radical production and destruction as well as their cycling, have to be taken into account; (2) losses of radicals on the walls in the experimental setup and inside the instrument have to be considered; and (3) potential interferences with isotopes of closed-shell molecules can impede quantification of detected $RO_2$ species. For example, an isotope of pinonic acid ($m/z$ 203.148 in $NH_4^+$ CIMS) strongly overlaps with OH-derived $RO_2$ species formed via oxidation of $\alpha$-pinene ($m/z$ 203.152 in $NH_4^+$ CIMS).

9. **Line 259 and fig.8: Higher oxidized $RO_2$ radicals arising from pure autoxidation steps show a mass difference of 32 mass units due to step-by-step insertion of molecular oxygen. A mass difference of 16 mass units points to efficient bimolecular $RO_2$ steps altering the autoxidation-governed $RO_2$ distribution. So, as already said, it would be fine to have the complete reaction conditions to get an idea how important $RO_2$ + $RO_2$ could be.**

The reviewer raises an interesting point. We agree that having a more complete understanding of the importance of $RO_2$ self-reactions could be beneficial for this study. However, to the best of our knowledge, kinetics of autoxidation and self-reactions is well studied for smaller $RO_2$ species only. Hence, we believe that determining the relative importance of chemical loss channels for $RO_2$ species lies beyond the scope of this study.

**References:**

Berndt, T., Richters, S., Jokinen, T., Hyttinen, N., Kurtén, T., Otkjær, R.V., Kjaergaard, H.G., Stratmann, F., Herrmann, H., Sipilä, M., Kulmala, M., and Ehn, M.: Hydroxyl radical-induced formation of highly oxidized organic compounds, Nature Communications, 7, 13677, DOI: 10.1038/ncomms13677, 2016.

Drozd, G.T., Kroll, J.H., and Donahue, N.M.: 2,3-Dimethyl-2-butene (TME) Ozonolysis: Pressure Dependence of Stabilized Criegee Intermediates and Evidence of Stabilized Vinyl Hydroperoxides, J. Phys. Chem. A 2011, 115, 161–166, DOI: 10.1021/jp108773d, 2011.

Newland, M.J., Rickard, A.R., Alam, M.S., Vereecken, L., Munoz, A., Rodenasd, M. and Bloss, W.J.: Kinetics of stabilised Criegee intermediates derived from alkene ozonolysis: reactions with $SO_2$, $H_2O$ and decomposition under boundary layer conditions. Phys. Chem. Chem. Phys., 17, 4076-4088, DOI: 10.1039/C4CP04186K, 2015.

Nguyen, T.B., Tyndall, G.S., Crounse, J.D., Teng, A.P., Bates, K.H., Schwantes, R.H., Coggon, M.M., Zhang, L., Feiner, P., Milller, D.O., Skog, K.M., Rivera-Rios, J.C., Dorris, M., Olson, K.F., Koss, A.R., Wild, R.J., Brown, S.S., Goldstein, A.H., de Gouw, J.A., Brune, W.H., Keutsch, F.N., Seinfeld, J.H., and Wennberg, P.O.: Atmospheric fates of Criegee intermediates in the ozonolysis of isoprene. Phys. Chem. Chem. Phys., 18, 10241-10254, DOI: 10.1039/C6CP00053C, 2016.

---

## Author Comment (AC2) · 9 Jan 2021

**Response to reviewer #2's comments**

Reviewer comments are in **bold**. Author responses are in plain text. Excerpts from the manuscript are in *italics*. Modifications to the manuscript are in *blue italics*. Page and line numbers in the responses correspond to those in the original AMTD paper.

**This study presents the development of an online method for measurements of SCIs and RO₂ in laboratory experiments using chemical derivatization and spin trapping techniques combined with H₃O⁺ and NH₄⁺ chemical ionization mass spectrometry. Application of this method is demonstrated using laboratory ozonolysis experiments of multiple hydrocarbons including TME, isoprene, pentene, hexene, alpha-pinene and limonene. The detection limits of spin trap and chemical derivatization agent adducts are estimated to be $1.4 \cdot 10^7$ molecule $cm^{-3}$ for SCIs and $1.6 \cdot 10^8$ molecule $cm^{-3}$ for RO₂ for 30 s integration time for the instrumentation used in this study. This manuscript is well written and within the scope of the journal. I recommend this manuscript to be published in AMT after the following issues be addressed.**

We would like to thank the reviewer for the positive reception of our work and constructive comments that helped us to improve our manuscript. Below we provide our replies to the reviewer's comments. Page and line numbers in the responses correspond to those in the AMTD paper.

1. **Page 6, Line 166-167: Is there any evidence for using HFA with SCIs to prevent secondary reactions?**

   HFA was used to study kinetics of various SCIs in the past (e.g., Drozd et al., 2011; Drozd and Donahue, 2011). In these studies, HFA was implemented to directly probe SCI formation.

   We modify the following paragraph by specifying that HFA was used to prevent SCI secondary reactions (P4 L109):

   *SCIs are known to be highly reactive towards ketones, especially electron poor ones such as HFA (Horie et al., 1999;  Taatjes et al., 2012). HFA has been previously used to effectively scavenge SCIs and prevent their secondary chemistry to directly probe SCI formation (Drozd et al., 2011; Drozd and Donahue, 2011).*

2. **Page 6, Line 191-194 and Page 8, Line 249-251: Could the author give more detailed explanations or quantitative analysis for these four reasons?**

   We add the following details to our description (P6 L191):

   *This discrepancy can be explained by a combination of the following factors. First, a fraction of (CH₃)₂COO·HFA adducts might be irreversibly deposited on the surfaces inside the experimental setup and the PTR 8000 instrument (Pagonis et al., 2017).  In addition, the sensitivity of observed SCI·HFA adducts depends on the reaction rate constant of the adduct with H₃O⁺ ion and the degree of fragmentation of protonated product ions SCI·HFA·H⁺ (Yuan et al., 2017). Since the reaction rate constant of SCI·HFA with H₃O⁺ ions is unknown, we assumed that all SCI·HFA adducts were*

*ionized via proton transfer from hydronium ions and therefore used the sensitivity we obtained from acetone calibration to quantify detected SCI·HFA species. In addition, we did not take into account possible fragmentation of SCI·HFA·H$^+$ ions which may impede their detection, although a first bond cleavage would likely only break the ozonide ring structure without loss of mass.*  *Finally, uncertainty of the kinetic model output is determined by the uncertainty in the SCI yield, and unimolecular and bimolecular reaction rate coefficients*

We add the following details to our description (P8 L248):

*Similar to experiments described in Sect. 3.1,* *several factors can contribute to this discrepancy: (1) gas-wall partitioning of RO$_2$ species and RO2·DMPO adducts in the* *experimental setup*  *and inside the PTR3 instrument; (2) uncertainty in sensitivity at which RO$_2$·DMPO adducts were detected; (3) potential fragmentation of RO$_2$·DMPO·NH$_4$$^+$* *product ions*; *and (4) uncertainties in the reaction rate coefficient $k_{RO_2+DMPO}$.*

3. **Page 9, Line 277-278: Since there could be various RO$_2$ in ambient air, how does the author think about the feasibility of using the CID technique to measure ambient air?**

   The CID technique can be used to constrain the instrument sensitivity to compounds that cannot be calibrated directly, including dozens of oxygenated compounds that were produced during a photooxidation experiment in an environmental chamber (Zaytsev et al., 2019). While we plan to implement analytical techniques presented in this study for ambient measurements of atmospheric radicals in the future, we think that these experiments are out of the scope of the current work.

4. **Supplement page 8: In Figure S11, at the beginning of the period DMPO+O$_3$, why did the SCI adduct (m/z 315.228) get a little increasing?**

   There are two factors that could contribute to the increase of SCI·DMPO tracer (*m/z* 315.228) when DMPO and O$_3$ were present in the experimental setup:
   1. formation of an isomer with same molecular formula but potentially different structure
   2. change in humidity of sampled air which affects both primary ion signal and sensitivity to observed compounds. As one can notice, other tracers (e.g., C$_{16}$H$_{27}$NO·NH$_4$$^+$, *m/z* 154.160; C$_{16}$H$_{26}$NO$_5$·NH$_4$$^+$, *m/z* 330.216) also showed a little increase when ozone was introduced in the experimental setup.

5. **Page 8, Line 234 and Supplement page 2, Line7: The last two letters of the word "CH$_3$C(=O)CH$_2$OO" use two different fonts.**

   We thank the reviewer for spotting this typo and fix it in the revised manuscript.

**References:**

Pagonis, D., Krechmer, J. E., de Gouw, J., Jimenez, J. L., and Ziemann, P. J.: Effects of gas–wall partitioning in Teflon tubing and instrumentation on time-resolved measurements of gas-phase organic compounds, Atmos. Meas. Tech., 10, 4687–4696, DOI: 10.5194/amt-10-4687-2017, 2017.

Yuan, B., Koss, A.R., Warneke, C., Coggon, M., Sekimoto, K., and de Gouw, J.A.: Proton-Transfer-Reaction Mass Spectrometry: Applications in Atmospheric Sciences, Chem. Rev., 117, 13187–13229, DOI: 10.1021/acs.chemrev.7b00325, 2017.